# Systematic Review: Contribution of the Gut Microbiome to the Volatile Metabolic Fingerprint of Colorectal Neoplasia

**DOI:** 10.3390/metabo13010055

**Published:** 2022-12-30

**Authors:** Robert van Vorstenbosch, Hao Ran Cheng, Daisy Jonkers, John Penders, Erik Schoon, Ad Masclee, Frederik-Jan van Schooten, Agnieszka Smolinska, Zlatan Mujagic

**Affiliations:** 1Department of Pharmacology and Toxicology, Maastricht University, 6227 AP Maastricht, The Netherlands; 2NUTRIM, School of Nutrition & Translational Research in Metabolism, Maastricht University, 6227 AP Maastricht, The Netherlands; 3Division of Gastroenterology-Hepatology, Maastricht University Medical Center, 6229 HX Maastricht, The Netherlands; 4GROW, School for Oncology and Developmental Biology, Maastricht University, 6227 AP Maastricht, The Netherlands; 5Department of Medical Microbiology, Infectious Diseases and Infection Prevention, Maastricht University Medical Center, 6227 AP Maastricht, The Netherlands; 6Department of Gastroenterology and Hepatology, Catharina Hospital, 5623 EJ Eindhoven, The Netherlands

**Keywords:** colorectal cancer, volatile organic compounds, microbiota

## Abstract

Colorectal cancer (CRC) has been associated with changes in volatile metabolic profiles in several human biological matrices. This enables its non-invasive detection, but the origin of these volatile organic compounds (VOCs) and their relation to the gut microbiome are not yet fully understood. This systematic review provides an overview of the current understanding of this topic. A systematic search using PubMed, Embase, Medline, Cochrane Library, and the Web of Science according to PRISMA guidelines resulted in seventy-one included studies. In addition, a systematic search was conducted that identified five systematic reviews from which CRC-associated gut microbiota data were extracted. The included studies analyzed VOCs in feces, urine, breath, blood, tissue, and saliva. Eight studies performed microbiota analysis in addition to VOC analysis. The most frequently reported dysregulations over all matrices included short-chain fatty acids, amino acids, proteolytic fermentation products, and products related to the tricarboxylic acid cycle and Warburg metabolism. Many of these dysregulations could be related to the shifts in CRC-associated microbiota, and thus the gut microbiota presumably contributes to the metabolic fingerprint of VOC in CRC. Future research involving VOCs analysis should include simultaneous gut microbiota analysis.

## 1. Introduction

Colorectal cancer (CRC) and its precursor lesions are associated with changes in metabolic pathways, as has been observed for several human biomaterials. These include volatile organic compounds (VOCs) in exhaled breath, feces, urine, blood, and tissue, which have received rising interest for their potential as non-invasive biomarkers. Gut microbiota perturbations are strongly associated with colorectal neoplasia, particularly CRC, and as a consequence of its activity, changes in VOC profiles may arise. However, to what extent the VOCs associated with colorectal neoplasia can be traced back to the gut microbiota activity is poorly understood. A better insight in these metabolic pathways is crucial to understand the origins of VOCs and thus their implications for future study designs aiming at the detection, monitoring of, and therapeutic interventions for colorectal neoplasia.

### 1.1. Colorectal Neoplasia

CRC is the third most common cancer worldwide with significant mortality, morbidity, global burden, and economic impact [1,2,3]. The heritability of CRC ranges between 12 and 35%, but the majority occurs sporadically [4]. In 70–90% of the sporadic cases, CRC develops from colorectal adenomas through the adenoma–carcinoma sequence, while sessile serrated lesions give rise to 10–20% of sporadic cases via the serrated pathway. Although these sporadic cases have a certain genetic predisposition, they are also strongly linked with environmental and lifestyle factors [5,6,7]. Smoking, obesity, decreased physical activity, consumption of alcohol, and red or processed meat have been associated with increased CRC risk. Likewise, exercise and high fiber intake are associated with a decreased risk of CRC. Different mechanisms underlie these associations, such as genotoxicity, inflammation, and immune regulation, and they may also involve the gut microbiota. All of these mechanisms could be reflected in any VOCs associated with CRC [8,9,10]. 

Early endoscopic removal of precursor lesions reduces the incidence of CRC and consequently leads to a significant reduction in CRC-related morbidity and mortality [11,12]. The fecal immunochemical test (FIT) commonly used in national screening programs comes with a considerable number of false positives and negatives [13]. Taking into consideration that colonoscopies are invasive procedures not without risks and their significant economic impact, a more accurate, non-invasive diagnostic tool for both CRC and (advanced) adenomas is highly sought after. In this respect, VOCs have been shown to be promising biomarkers [14].

### 1.2. Volatile Organic Compounds 

First identified by Pauling et al. in 1971, VOCs are volatile carbon-based end products of host and microbial metabolism [15,16]. Variation in VOC profiles results from external factors such as diet, lifestyle, medication usage, and environmental factors and host-related factors such as disease state, genetics, and the gut microbiome. Over 1800 VOCs have been identified in healthy individuals using the gold standard gas chromatography–mass spectrometry (GC–MS), and some resulted in clinical utility, such as urea breath testing for the detection of *Helicobacter pylori* infection and the recently FDA-approved breath test for COVID-19 [17,18,19]. A large proportion of all volatiles (or the volatolome) is produced by the gut microbiome via fermentation and enzymatic reactions, resulting in short chain fatty acids (SCFA), proteolytic fermentation products (e.g., branched chain fatty acids (BCFA), phenols, sulfides), ketones, and alcohols amongst others [20]. The strong link between VOCs and the gut microbiota composition has been underlined by Smolinska et al., who found significant correlations between VOC profiles in exhaled breath and the abundance of specific fecal bacteria during active and inactive stages of inflammatory bowel disease [21].

The shifts in CRC-associated gut microbiota composition and activity might translate into measurable outcomes in the human VOC profile. Several studies have identified altered VOC profiles associated with CRC and, to a lesser extent, colorectal adenomas, when compared to (healthy) controls [22,23,24]. VOCs of interest for CRC detection have previously been categorized as those originating from altered cell energetics, alterations in enzymatic activity of cancerous cells, and oxidative stress in a review by Monedeiro et al. [25]. However, the process of VOCs arising due to shifts in the gut microbiota was only briefly mentioned in their review. Zhang et al. reported a detailed mechanistic overview of gut microbiota-derived metabolites in CRC but limited their overview to SCFAs, bile acids, and tryptophan metabolites [26]. Taken together, the influence of the gut microbiota on the human volatolome is often not considered in metabolomics studies aiming to detect colorectal neoplasia. Therefore, in an attempt to elucidate the potential origin of CRC-associated VOC profiles, this systematic review aims to provide an overview of the changes in the human volatolome in colorectal neoplasia (i.e., CRC and colorectal adenoma) by relating biologically relevant VOCs to CRC-associated gut microbiota. 

First, a systematic search was performed on studies that have analyzed VOCs in various biological matrices for the detection of colorectal neoplasia in patients as compared to controls. This was complemented with a second search that considered systematic reviews on CRC-associated gut microbiota composition. These combined data demonstrate a complex interaction between the human volatolome and CRC-associated gut microbiota composition. Finally, the implications of this knowledge on technical and methodological aspects of VOC research, what current knowledge gaps exist, and suggestions for future research directions are elaborated. 

## 2. Materials and Methods

### 2.1. Search Strategy 

A systematic search of electronic databases including PubMed, Embase, Medline, Cochrane Library, and the Web of Science was performed until February 2022 without any restrictions. This systematic review was conducted using the Preferred Reporting Items for Systematic Reviews and Meta-Analyses (PRISMA) guidelines [27]. First, the following terms were searched using the Boolean operators OR “volatile organic compound”, “volatolome”, “volatolomics”, “breathomics”, “gas-chromatography”, and “gas-chromatography mass-spectrometry” combined with the Boolean operator AND with “colorectal carcinoma”, “colorectal cancer”, “colon cancer”, “rectal cancer”, “colorectal tumor”, “colorectal neoplasia”, “colorectal neoplasm”, “colorectal polyps”, and “colorectal adenoma”.

Second, CRC-associated gut microbiota compositions were systematically searched using the terms “microbiome”, “microbiota”, “gut microbiome”, and “gut microbiota”. This search was combined with the terms used for colorectal neoplasia in the first search using the Boolean operator AND. Only systematic reviews were considered due to the large number of publications on CRC-associated gut microbiota in recent years. 

Further details on the search strategies can be found in the Appendix A (i.e., Appendix A). 

### 2.2. Selection Criteria

Two authors (RV and HC) independently reviewed titles and abstracts of all retrieved articles. Human studies were included if they specified VOC identification in any biological matrix in relation to sporadic colorectal neoplasia compared to controls (i.e., subjects without colorectal neoplasia). Sensor-based applications were excluded as they do not allow for VOC identification. The PRISMA flowchart is shown in Figure 1a. The search yielded a total of 3156 records, of which 1474 were duplicates. Then, 1682 abstracts were independently screened by HC and RV. Reviews were scrutinized for missed publications. In case no consensus was reached, the article was retained for the next selection phase. Only full text articles of original studies in English were included, leading to the exclusion of reviews, abstracts, letters, editorial, and non-English publications (n = 1504). Full texts of twelve articles could not be retrieved and were excluded. The full texts of 166 studies were assessed for eligibility. Then, 99 studies were excluded for not meeting the inclusion criteria. Snowball referencing of articles in reviews yielded an additional ten potential studies of which seven were excluded. One study conducted by the authors was accepted for publication at the time of writing and was included for analysis. A total of 71 studies were included for analysis. 

A second systematic search on CRC-associated gut microbiota is shown in Figure 1b. The search yielded a total of 14,126 records. After filtering for systematic reviews, meta-analyses (n = 391), and duplicates removed (n = 159), a total of 232 records were independently screened. Systematic reviews that provided data on gut microbiota composition in colorectal neoplasia compared to controls in any biological matrix were included in the review. Reviews targeting specific microbes and animal studies were excluded. Only full text articles of systematic reviews and meta-analyses in English were included, leading to the exclusion of abstracts, letters, editorials, original studies, and non-English publications (n = 189). The full text of one article could not be retrieved and was excluded. The full texts of 45 studies were assessed for eligibility. Five systematic reviews were included for data extraction.

### 2.3. Data Extraction 

The following data was extracted per study: author, year of publication, study design, study population, number of subjects, CRC or precursor lesion, biological matrix, platform for chemical analysis, individual discriminatory VOCs to detect diseased versus controls, relative changes in their concentrations and their statistical significance, and, if provided, associated microbiota activity and/or abundancies. The sensitivity, specificity, and accuracy of VOC profiles were not extracted due to high variation of selected VOCs combined with a general lack of validation efforts. In case VOCs were repeatedly reported by various studies as discriminatory, but with varying higher and lower concentrations toward the colorectal neoplasia group, the overall change in concentration reported here was defined as by the majority. We further distinguish between consistent and inconsistent study outcomes (i.e., studies showing similar or contradicting changes in VOC concentrations, respectively). In case relative changes were not reported or when there was a tie between studies reporting higher or lower concentrations in the colorectal neoplasia group, their change was reported as undetermined. 

For the second part of the review, reported changes in gut microbiota abundancies on genus and species level in colorectal neoplasia cases as compared to controls were extracted. Only changes that were present in both adenomas and either CRC or CRC were extracted. If authors did not provide details on the mechanisms of CRC-associated microbiota on colorectal carcinogenesis (and therefore, possibly VOC formation) the additional literature was consulted. The results of the first search were then interpreted in light of the second search.

### 2.4. VOC Data Interpretation 

Interpretation of the extracted VOC data is not straightforward. Chemical analyses are subject to inclusion and exclusion biases depending on the chemicals of interest and the nature of the sampling, separation, and detection platforms used. Moreover, studies may be exploratory (i.e., untargeted) or hypothesis-driven (i.e., targeted). As a consequence, studies are often not directly comparable due to the large heterogeneity in methodologies, blurring the boundaries of reproducibility. In addition, the variation in clinical study designs, data-analytical approaches, and the inherent nature of the biological matrix in which VOCs can be detected further add to the heterogeneity in outcomes between studies. Therefore, an interpretation that respects expected variation in study outcomes but preserves relevant information of the biological system as a whole is needed. To be able to generalize observations between studies, VOCs were subdivided into biological or chemical classes in which VOCs were expected to show similar behavior. VOCs that were reported only once and did not fit any of these groups, or with complex molecular structures (e.g., highly branched with many sub-groups) and that had an unverified identity (i.e., using standards), were excluded from this review. VOCs that were observed more than once were included for further analysis. VOCs that were reported only once within a specific biological matrix but did fit these defined groups as observed in other biological matrices were included.

## 3. Results

### 3.1. Description of VOC Studies 

Included studies were published between 2008 and 2022 and consisted of 71 original studies [28,29,30,31,32,33,34,35,36,37,38,39,40,41,42,43,44,45,46,47,48,49,50,51,52,53,54,55,56,57,58,59,60,61,62,63,64,65,66,67,68,69,70,71,72,73,74,75,76,77,78,79,80,81,82,83,84,85,86,87,88,89,90,91,92,93,94,95,96,97,98] (Figure 1a). The majority (n = 68) analyzed CRC cases compared to controls and eight studies included adenomas [33,36,45,68,69,96,97,98]. VOCs were analyzed in several biological matrices, including feces (n = 11) [31,33,36,49,52,63,87,89,90,92,98], urine (n = 9) [34,35,62,65,74,75,78,80,82], breath (n = 10) [28,29,40,51,60,61,70,72,85,96], blood (n = 21) [37,38,43,44,45,46,48,53,56,57,66,67,76,79,81,83,84,86,88,91,95], tissue (i.e., as cell lines or resected intestinal tissues) (n = 17) [32,39,42,47,50,54,55,58,59,64,68,69,73,77,93,94,97], saliva (n = 1) [30], tissue and breath (n = 1) [41], and tissue and feces (n = 1) [71]. Seven studies also included data on fecal microbiota composition and/or activity [33,36,52,68,87,89,90,92] and one study on tissue [68]. Confounders, most importantly diet, medication usage, and other lifestyle factors, were not considered by the included studies unless explicitly stated. 

Based on the reported VOCs, the following chemical or biological classes were defined: SCFAs, proteolytic fermentation products (including BCFAs and sulfides), amino acids, bacterial fermentation products, (branched) alkanes/enes, VOCs related to Warburg and tricarboxylic acid cycle (TCA) metabolisms, aldehydes, ketones, furans, and terpenes. We acknowledge BCFAs, sulfides, and, to some extent, amino acids do result from proteolytic fermentation, but due to their different chemical properties they were categorized into these classes. VOCs belonging to the class here termed bacterial fermentation should be regarded as the general metabolic output of the microbiome, not specifically related to either fiber or protein fermentation. A comprehensive overview of the obtained VOCs and their trend per matrix is shown in Table 1. A brief summary per matrix on study methodologies and reported marker VOCs is provided below.

### 3.2. Fecal VOC Profile 

Twelve studies analyzed fecal VOCs for the detection of CRC (Appendix A). The VOC sampling of fecal headspace was performed either using solid phase microextraction (SPME; n = 1) or using bags (Nalophan n = 1; Tedlar n = 1) with the subsequent transfer to thermal desorption (TD) tubes, or directly from the feces itself using derivatization methodologies (n = 9). The sample size ranged between 21 and 688 subjects. One study analyzed VOCs using selected ion flow tube mass spectrometry (SIFT-MS), one study used capillary electrophoresis–time of flight–mass spectrometry (CE–tof–MS); all others used GC–MS. 

The most frequently reported differences between CRC cases and controls were among SCFAs, BCFAs, and amino acids. SCFAs acetic acid, propionic acid, and butyric acid were most frequently reported with a majority of studies reporting decreased concentrations in CRC cases [33,36,52,64,87,90,92]. Increasing concentrations were observed for BCFAs, mostly for isovaleric acid and isobutyric acid [33,63,87]. Amino acids were consistently reported in higher concentrations in CRC cases. These findings coincided with increased concentrations for other proteolytic fermentation products, including phenolic compounds, cadaverine, hydrogen sulfide, and ammonia. Compounds that were linked to the TCA cycle were frequently reported, but the overall pattern remained ambiguous. Succinate, oxalate, malate, and glutamate were found to be increased [52,87,89], whereas maltose and fructose were found in decreased concentrations in diseased populations [71,90]. Other reported outcomes for fecal VOC profiles include increases in aldehydes [98], lactic acid [89], and terpenes [31] in CRC cases. 

### 3.3. Dynamics between Fecal VOCs and Gut Microbiota

Seven studies simultaneously studied fecal microbiota composition and/or activity as well as fecal VOCs [33,36,52,87,89,90,92]. The results obtained in these studies can be subdivided into three categories: (1) significant differences between microbes/VOC concentrations within a study design that takes confounders into account, (2) coincident findings of marker VOCs and marker microbes, and, lastly, (3) correlating patterns of microbial abundancy/activity with VOC concentrations (i.e., considering only within disease group correlations).

Only Chen et al. simultaneously considered dietary effects in addition to microbiota abundancy and fecal VOCs in advanced adenomas (AA) [33]. They observed reduced butyrate concentrations for healthy subjects with low-fiber diets compared to those with high-fiber diets. Similarly, reduced butyrate concentrations were observed in AA cases compared to controls. Interestingly, butyrate concentrations for AA cases did not differ between high- and low-fiber diets while AA cases with high fiber diets showed significantly lower amounts of butyrate compared to high fiber controls. These observations coincided with lower abundancies of *Clostridium* spp., *Roseburia* spp., and *Eubacterium rectale*, supposedly butyrate producers. Altogether, these observations suggest altered long-term dynamics in the gut microenvironment that inhibit butyrate production from dietary fibers. Other studies did not consider dietary effects but underlined changed dynamics of the gut microbiome. Kim et al. reported coinciding patterns of increased alanine, leucine, and isoleucine concentrations with increased Firmicute abundancy in CRC cases compared to controls [52]. Similarly, decreasing concentrations were shown for lysine, tyramine, aminobutyric, ethanol amine, phenol, and butyrate. Opposite associations were reported for Pseudomonadota (formerly Proteobacteria) and Actinomycetota (formerly Actinobacteria). Similarly, Weir et al. observed Bacillota (formerly Firmicutes), Bacteroidota (formerly Bacteroidetes), Pseudomonadota, and *Verrucomicrobia* abundancies coincide with increased amino acid concentrations [87]. 

Other studies correlated microbial abundancies to VOC patterns and found these to be disease-stage specific. Coker et al. reported correlations of *Peptostreptococcus anaerobius* with glycine, *Parvimonas micra* with valine, *Clostridium symbosium* with valine and homoserine, *Synergistes sp.* with aspartic acid and tyrosine, *Porphyromonas gingivalis* with gamma-aminobutyric acid, and *Prevotella nigrescens* with asparagine [36]. These correlations were either stronger or weaker upon CRC progression. In addition, increased amino acid concentrations correlated to decreased butyrate concentrations. The changed dynamics of these amino acid concentrations occur early in the onset of disease before adenomas become cancerous, suggesting that this metabolic state may cause susceptibility for further disease progression. Yachida et al. came to similar conclusions. They demonstrated that *Atopobium parvulum* (a known hydrogen sulfide producer) and *Actinomyces odontolyticus* are significantly increased in early-stage CRC, but not in later-stage CRC. Meanwhile, amino acids concentrations in AA cases were higher compared to controls [89]. The relationships above, based on microbial abundancies, were further underlined by their observations of microbial activity, either via up- or down-regulated pathways [89]. They observed the upregulation of aromatic amino acids, including phenylalanine and tyrosine, and sulfide-producing pathways were associated with CRC. Indeed, increased sulfide concentrations are observed in fecal VOC studies. Tryptophan-producing pathways were depleted, which coincides with findings by Coker et al. who found altered pathway abundancies for these and nitrogen-containing amino acid productions [36]. Other interesting mechanic insights were observed by Yusuf et al., who demonstrated taxa-specific differences within the genus *Bifidobacterium* between CRC and healthy controls [92]. Combined, these results suggest a difference in long-term gut dynamics of the microbiome. However, few studies sufficiently included a multi-omics strategy while considering important confounders such as diet. 

### 3.4. Urinary VOC Profile 

Nine studies analyzed VOCs in urine (Appendix A). The sample size ranged between 96 and 274 subjects. Four studies analyzed urine samples via derivatization techniques, all others focused on urinary headspace, either via direct headspace injection, SPME, or TD tubes.

The variation in outcomes between studies was large. The majority of urinary VOCs was reported only once and for many VOCs their relative increasing or decreasing concentrations in CRC cases were either not reported or often inconsistent between studies. SCFAs were not reported as markers. In general, BCFAs [35,78] as well as amino acids [34,35,65,75] were observed in higher concentrations in CRC cases. In line with BCFAs, levels of phenol and sulfides, and other products of proteolytic fermentation were higher in most of the studies [35,74,75,80,82]. Aldehydes, including butanal, octanal, and decanal were increased in CRC cases [74,82]. Branched alkanes/-enes, complex benzenes, naphthalenes, and cyclohexenes/-anes, were reported in higher concentrations in CRC cases. Although this general pattern was confirmed in multiple studies, the chemical identities of VOCs did not match between studies. VOCs related to the TCA cycle were generally present in decreased concentrations, with the exception of glucose, fumarate, and glutamate, which were found in higher concentrations in CRC cases. Other observations included increased concentrations of terpenes and furans. Lastly, lactic acid was found in higher concentrations in CRC cases [34].

### 3.5. Exhaled Breath VOC Profile 

Eleven studies analyzed VOCs in exhaled breath (Appendix A) with sample sizes ranging between 27 and 382 subjects. Breath collection methodologies varied from the usage of sampling bags (n = 7, Nalophan, n = 1; Tedlar, n = 4; Mylar, n = 2) with the subsequent transfer to TD tubes, syringe extractions (e.g., bio-VOC sampling or SPME, n = 2), to next-generation devices (ReCIVA, n = 2) that directly capture VOCs on TD tubes without the use of bags. The chemical analysis consisted of SIFT–MS, ion molecule reaction–mass spectrometry (IMR–MS), and GC–MS (n = 8). 

Despite studies describing positive associations between breath VOC profiles and CRC, a shared presence of associated individual VOCs in CRC was hardly observed. VOCs reported to be discriminative most often and in increased concentrations were alkanes, aldehydes, and acetone, but the majority of metabolites was only reported once. Moreover, even when VOCs were divided into chemical classes, common findings were rare. Compared to other matrices, some observations did show shared patterns that are worth addressing. Although only reported once, higher BCFA concentrations in breath in CRC cases were observed [51]. Other proteolytic fermentation products were detected [41,61] but trends were inconclusive as were they for terpenes [40]. Observations of SCFAs between studies were inconsistent, but they were repeatedly reported as important VOCs [28,41,72]. Similarly, lactic acid was raised in concentration in CRC cases [96]. No compounds related to the TCA cycle were reported. 

### 3.6. Blood VOC Profile 

Twenty-one studies analyzed VOC profiles in blood (Appendix A). Amongst these, a distinction was made between whole blood, plasma, and serum. As the VOC markers concluded from these studies did not differ between these matrices, this review does not further elaborate on this distinction. The sample size ranged from 10 to 574 subjects. Nineteen studies used derivatization techniques and two studies performed headspace analysis using SPME. For the subsequent chemical analysis, seventeen studies applied GC–MS, three studies performed GCxGC–MS, and one used capillary electrophoresis–mass spectrometry (CE–MS). 

Lactic acid was the most frequently reported marker and, together with pyruvic acid, was observed in consistently higher concentrations in CRC cases compared to the controls [43,44,46,48,67,76,81,91]. Other observations were lower and higher concentrations for SCFAs and BCFAs, respectively. Amino acids were largely present in decreasing concentrations toward the CRC group, although exceptions included increasing concentrations for alanine [44,48,66,83], glycine [44,48,57,65], and isoleucine [44,48,66,88]. VOCs related to the TCA cycle showed mainly inconclusive patterns, although, notably, 2-hydroxybutanoic acid, a known marker for disturbed energy metabolism, was observed in strongly increased concentrations [44,48,66,76,81,83]. Proteolytic fermentation products, aldehydes, and terpenes were not reported or rarely reported. 

### 3.7. Tissue VOC Profile 

Nineteen studies analyzed VOC profiles in intestinal tissue samples (Appendix A). Tissue samples were either taken from patients (biopsies or resection material), where cancerous tissue was compared to healthy tissue in a proximity of 5–10 cm or from cell lines. Fourteen studies combined tissue grinding with derivatization techniques, while five measured tissue headspace using SPME. As no clear differences were obtained between these methodologies, we do not further differentiate between them. The sample size ranged between 6 and 376 samples. Fourteen studies performed chemical analysis using GC–MS, one study used GCxGC–MS, and the remaining study performed GC–isotope ratio mass spectrometry. One study simultaneously analyzed microbiota abundancies [68].

Valeric acid and BCFAs were present in higher concentrations in CRC samples, although these observations were rare as were they for other proteolytic fermentation products. Amino acids and lactic acid were observed to be present in higher concentrations in CRC samples [58,59,64,77,93]. Other chemical groups consisted of aldehydes, although their change in concentration remained ambiguous or unreported, and ketones, which were generally present in decreased concentrations in CRC cases. Likewise, VOCs related to the TCA cycle were generally present in lower concentrations. As observed by Nugent et al., microbiota abundancies coincided with volatiles relating to the TCA cycle and amino acid concentrations [68]. These included *Lactobacillus* sp., *Escherichia coli*, *Bifidobacterium* sp., *Clostridium* sp., *Bacteroides* sp., and *Eubacteria* [68].

### 3.8. Saliva VOC Profile 

One study analyzed salivary VOC profiles and included 18 CRC cases and 16 healthy controls [30]. VOC sampling was performed by derivatization followed by GC–MS chemical analysis. Several short alcohols were reported to be present in deviating concentrations in CRC cases. Furthermore, ethanol and methanol levels were at higher concentrations while 1-propanol and 2-propanol levels were at lower concentrations in the diseased group. Additionally, increasing acetaldehyde and acetone concentrations with decreasing ethylacetate were observed. 

### 3.9. Summary of the Volatolome and Gut Microbial Microenvironment in Colorectal Neoplasia 

The majority of studies focused on CRC (n = 68), whereas only a minority included patients with colorectal adenomas (n = 8). No clear differences between the outcomes of these studies were observed, and recent studies have argued for the early onset metabolic changes, increasing susceptibility for disease progression [36]. The most frequently reported changes of volatiles over all matrices in patients with CRC were SCFAs, proteolytic fermentation products (i.e., BCFA, amino acids, and sulfide), and compounds relating to the Warburg and TCA metabolism. Typically, most consistent results were observed for fecal, tissue, and blood analyses, whereas urinary and breath studies showed more variation in outcomes. This variation could be (in part) related to the heterogeneity in methodologies used. Moreover, exhaled breath analyses typically show a higher degree of noise compared to other platforms. Despite some inconsistencies between matrices, it can be concluded that, for CRC cases, SCFA levels are generally decreased compared to non-CRC controls. Although most studies did not consider diet here as a confounder, one study did show that the disease effect was bigger than the diet effect. These findings coincided with increased concentrations of proteolytic fermentation products in the CRC group. BCFAs showed the most consistent results across all types of matrices with higher concentrations in CRC cases. Other consistent changes between controls and CRC cases were observed for lactic acid and pyruvic acid with increased concentrations over all biological matrices. Likewise, amino acids were reported in all matrices, although results were inconsistent. In general, increased concentrations were reported in feces, urine, and tissue samples, but decreased concentrations in blood. Only a few studies simultaneously analyzed VOCs as well as gut microbiota profiles. Notably, those that did observed amino acid concentrations to be especially strongly associated with microbiota abundancies and activity. Compounds related to the TCA cycle were generally reported to be present in lower concentrations in CRC cases. Other classes, such as furans, aldehydes, and terpenes, were all present in increasing concentrations toward CRC cases but not always reported in every matrix. Feces, breath, blood, and tissue would appear the most valuable matrices for VOC identification. Feces and blood may be most informative, with more consistent results for feces, but changes in TCA metabolism were additionally detectable in blood. Breath analyses suffer from large variations in outcomes between studies, but as the most patient-friendly platform, achieve the highest participation rates. Tissue samples are unsuitable for non-invasive screening strategies. However, they may provide insight into where VOCs originate. For these samples, the most obvious differences were among TCA-related metabolites and amino acids, suggesting a complex interplay between the microbiota and colonic tissue for at least these latter metabolites. 

In summary, these results indicate an increase in proteolytic fermentation and a decrease in saccharolytic fermentation in CRC cases.

### 3.10. CRC-Associated Gut Microbiota

Four systematic reviews and one systematic review with meta-analysis described data on microbial abundancies in colorectal neoplasia [99,100,101,102,103] (Figure 1b). An overview of the altered gut microbiota abundancies in colorectal neoplasia is provided in the Appendix A. Huybrechts et al. described microbiota on the risk of cancer. There were 124 studies until 2019 included, of which 50 studies linked microbial abundancies with CRC risk. Microbiota data analyses were performed using 16S rRNA and genome shotgun metagenomics in feces and mucosal biopsies. The outcome was the relative abundancy of taxa in cancer versus non-CRC controls, considering only the results that were shared with at least one other study. Amitay et al. reported a systematic review on the fecal microbiota in both CRC and adenomas between 2007 and 2017, and a total of 19 studies were included. Most studies performed 16S rRNA gene sequencing, while other analyses included qPCR or a combination of both. Microbes found in at least two studies to significantly differ in abundance (i.e., in the same or opposite direction) between cases and control groups were included. Borges-Canha et al. reported a systematic review with unclear inclusion criteria until 2014. There were 45 studies included. Both animal and human studies were included, and microbiota were analyzed in tissue and fecal samples. Only gut microbiota abundancies in human samples were extracted for this review. Liu et al. reported a systematic review and meta-analysis comparing gut microbiota in CRC versus healthy controls until 2015 and included five studies using qPCR on fecal samples. Aprile et al. is the only systematic review that analyzed intestinal microbiota in adenoma patients compared to controls from 2010 until 2020, and 19 studies were included. Analysis was performed on fecal and intestinal mucosa biopsies. Analyses were performed using qPCR, whole metagenome shotgun sequencing (WMGS), and 16S rRNA gene amplicon sequencing.

The methods used for the systematic reviews on CRC-associated gut microbiota varied widely, and not all reviews had clear inclusion criteria [100,103]. In general, microbes that have been associated with chronic inflammation, DNA damage, and immune regulation, such as the genera *Porphyromonas, Peptostreptococcus,* and *Leptotrichia* and species *Fusobacterium nucleatum*, *Parvimonas micra,* and *Bacteroides fragilis,* were abundant in CRC and adenoma cases when compared to controls. The genera *Bifidobacterium*, *Roseburia*, and *Eubacterium* and species *Faecalibacterium prausnitzii* (known SCFA producers) were consistently found to be depleted in CRC cases [99,100,101,102,103]. Interestingly, the genera *Atopobium* and *Desulfovibrio* (associated with hydrogen sulfide production) were increased in CRC cases in two reviews [99,101]. Other genera, such as *Gemella*, *Odoribacter*, *Slackia*, *Colinsella*, *Dysgonomonas*, and *Selenomonas*, were found in increased abundancy in CRC cases but were only reported once. A more detailed overview of up- or down-regulated microbes, reported at genus or species level, is presented in Figure 2. Although microbiota perturbations can be a consequence of neoplasia presence, they can also contribute to neoplastic development and/or progression. Briefly, gut dysbiosis may promote CRC through various processes, including chronic inflammation or immune response and the biosynthesis of toxic metabolites [104]. Gut microbiota-induced tumor-promoting inflammation is characterized by the production of cytokines by resident innate immune cells and the establishment of an immunosuppressive tumor microenvironment (TME). This can compromise the intestinal barrier function and may trigger a further influx of pathogenic microbes and/or their metabolites from the intestinal lumen and contact with intestinal epithelial cells (IEC) [26]. These microbes may elicit colorectal carcinogenesis by direct DNA damage, genomic instability, proliferative signaling, apoptosis, and chronic inflammation, which are described in detail elsewhere [105]. For example, the direct damage of DNA is caused by the colibactin production by *pks^+^ Escherichia coli* or the bacteroides fragilis toxin by *Bacteroides fragilis*, and proliferative pathways may be activated by enabling the Wnt/β-catenin signaling pathway via *Fusobacterium nucleatum* through FadA binding to host E-cadherin. Notably, *pks^+^ Escherichia coli* has been until now the only identified microbe that has a direct causal link to colorectal carcinogenesis [106]. 

## 4. Discussion

Changes in VOC profiles associated with colorectal neoplasia have been observed in all human biological matrices. The current systematic review provides an overview of the changes in the human volatolome in colorectal neoplasia by considering biologically relevant VOCs, not only individually but also according to their biological classes and in light of CRC-associated gut microbiota abundancies as visualized in Figure 2 and further discussed below.

### 4.1. SCFAs and Proteolytic Fermentation Products

SCFAs are volatile fatty acids produced mainly by gut microbiota in the (proximal) colon from dietary polysaccharides [107]. The most prevalent SCFAs are acetate, propionate, and butyrate, and they are generally known for their beneficial effects. They are an important source of energy for intraepithelial cells (IEC) and can affect gut motility and improve IEC integrity by promoting tight junctions, epithelial cell proliferation, and host-microbe signaling, amongst others [108]. SCFAs can be produced in relatively large or low quantities depending on dietary intake and the microbiota composition, together contributing to either saccharolytic or proteolytic fermentation, respectively. Intermediates that are involved in these conversions include succinate, lactate, or acetyl-CoA [109]. These are metabolized into SCFAs by the enzymes of *Bifidobacterium* spp., *Roseburia* spp., *Ruminococcus* spp., and *Faecalibacterium prausnitzii*, amongst others, through several pathways and microbe–microbe cross feeding (see Figure 2) [110]. Interestingly, coinciding with a shift toward proteolytic fermentation, these beneficial SCFAs and their producers were observed to be present in decreased concentrations and abundancies in CRC cases, respectively (Figure 2). Accordingly, increased concentrations of VOCs related to proteolytic fermentation have been observed. Here, BCFAs demonstrated the most consistent results. These are exclusively produced by the fermentation of branched chain amino acids. The health effects of BCFAs are not yet fully understood but they are generally linked to disruption of interstitial linkages between IECs, causing reduced intestinal barrier integrity (or ‘leaky gut’) and, consequently, the leakage of pathogen-associated molecular patterns into the bloodstream, triggering low-grade systemic inflammation [107,111]. Other proteolytic fermentation products include ammonia, amines, phenols, indoles, and sulfurous compounds [112]. The majority of these metabolites are thought to be harmful to intestinal health. Amino acids can be catabolized into amine by bacteria, such as bifidobacteria, clostridia, lactobacilli, enterococci, and streptococci [113]. The degradation of aromatic amino acids yields phenolic compounds. Exceptions to such detrimental effects are metabolites related to tryptophan catabolism, where *Bacteroides* and *Enterobacteriaceae* produce indoles. Indoles have been linked to enhanced host defense by increasing tight junction protein expression and downregulating the expression of pro-inflammatory cytokines [114]. In summary, these findings are in line with known dietary risk factors for CRC but it remains unclear whether these changes are a reflection of this risk or a representation of true pathohistological changes. Studies analyzing the direct relationship between VOCs and the gut microbiota have emphasized this decrease in saccharolytic fermentation and increase in proteolytic fermentation and the possible involvement of the gut microbiota herein. 

### 4.2. Amino Acids

Increased amino acid concentrations were observed in almost all matrices toward colorectal neoplasia. The decreased concentrations that were observed in blood could be explained by an increased influx of amino acids into cancerous cells to meet their increased protein synthesis and energy demands [115]. The typically Western protein-rich and low-fiber diet is associated with CRC and could be the cause or a confounder of the observed amino acid profiles [116]. Therefore, increased amino acid concentrations may result from dietary habits, gut microbiota shifts, or a combined effect. However, this exact interplay is not yet fully elucidated. Chen et al. did observe AA cases with similar fiber intake as healthy controls to excrete less butyrate, thereby demonstrating the inhibited ability of the microbiota to metabolize dietary fibers [33]. A possible explanation for such reduced fiber fermentation could relate to mucus degradation in the gut. The main structural and physical components of mucus are glycosylated proteins or mucins. A long-term diet low in fibers can shift the microbiota to metabolize host mucins resulting in the observed increased amounts of proline, serine, and threonine in CRC cases [117]. Furthermore, sulfur reducing bacteria metabolize sulfated glycans to hydrogen sulfide. Hydrogen sulfide is toxic to IECs, pro-inflammatory, and further facilitates mucin degradation. Indeed, fecal samples of CRC patients contained higher abundancies of *Akkermansia muciniphila,* a mucin-degrader, and sulfate-reducing genera such as *Atopobium, Bilophila,* and *Desulvofibrio.* Higher abundancies of the latter coincided with increased hydrogen sulfide concentrations (Figure 2) [118]. 

### 4.3. Tricarboxylic Acid Cycle and Warburg Metabolism 

A well-known phenomenon in cancerous cells is the decreased metabolic activity of the TCA cycle. Although the gut microbiota is not directly involved in this process, it may contribute to this process by providing alternative nutrients. Lactic acid, an SCFA, produced through fermentation reactions by the gut microbiota, can be converted into pyruvate, which can be utilized as an energy substrate via the Warburg metabolism. Moreover, lactic acid stimulates the TME via angiogenic effects [119,120]. Simultaneously, lactic acid concentrations can build up and be excreted by the cancerous cells themselves via the aerobic glycolysis of glucose. Altogether, this two-way mechanism results in the accumulation of a lactic acid pool, which influences both carcinogenesis as well as the microbial environment. 

### 4.4. Aldehydes and Ketones 

Aldehydes have mainly been reported as discriminatory VOCs in breath and urine with higher concentrations present in CRC cases. Aldehydes can be metabolized by the microbiota through the conversion of alcohols by, e.g., *Ruminococcus, Prevotella,* and *Bifidobacterium* [121], but also result as a generic product of oxidative stress [122]. Their increasing concentrations could be cause and consequence of the aforementioned effects: interrupted gut barrier integrity, increased (systemic) oxidative stress, and disturbed immune responses. Branched, more complex versions of aldehydes can result from oxidative stress of the microbiota as their cell membranes contain unsaturated branched chain fatty acids, although such VOCs can typically be hard to identify. Ketones arrive from similar processes, although they can also result from the detoxification of SCFAs in the liver [123]. 

### 4.5. Terpenes and Furans 

Several studies reported increased concentrations of terpenes in feces, urine, and breath toward CRC cases. Previous research has not directly related terpenes to microbial shifts in CRC. However, they have been related to pulmonary fungal infections [124,125]. Therefore, it is plausible that they might be linked to detrimental mycobiome profiles in CRC cases [126,127]. The gut virome and mycobiome have been associated with colorectal carcinogenesis [128,129]. However, very little is known about the functional signatures of fungal and viral communities [130,131]. Nonetheless, its potential contribution in colorectal carcinogenesis should not be forgotten [21]. Similarly, furans have not been related to CRC directly. Their toxic effects are acknowledged, although their long-term effects have not been fully elucidated. Their origin could lie within host Maillard reactions [132]. The involvement of both groups of compounds is highly speculative as their origin could also be derived from plant-based diets or smoking and coffee consumption, respectively [133,134]. 

### 4.6. Alcohols and Other Bacterial Fermentation Products

Alcohols can be produced via microbial fermentation reactions, such as ethanol and propanol. Pseudomonadota (formerly Proteobacteria), which are generally enriched in CRC patients, are known for their capability of alcohol generation [121]. In turn, alcohols can also be transformed into SCFAs by the liver through detoxification. Altogether, however, alcohols were not often among the most discriminating VOCs. Although they might be correlated to microbial activity, more research is needed to verify this and properly understand the effect of confounders such as alcohol consumption. 

## 5. Experimental Designs and Future Research 

One of the major challenges of the current systematic review is the inability to properly assess the reproducibility of results between studies due to the lack of standardization. Here, we refer to the standardization of not only metabolic measurements but also the data-analytical procedures involved as well as consideration of confounders and potential contaminants. The standardization of VOC sampling and analysis for colorectal neoplasia detection is essential before implementation in clinical practice becomes within reach [19]. Many studies reported insufficient details on their clinical design, while others valued the effects of fasting and bowel cleansing differently. The only clinical design that allows for the equal treatment of groups with respect to such effects while focusing on relevant distributions in a population is a prospective design where patients are sampled before bowel cleansing. If studies sample after bowel cleansing and under dietary constrictions, their effects (and duration of effects) need to be understood. Equally important, however, is elucidating the complex interplay between the gut microbiome, the volatolome, and the influence of external factors in colorectal neoplasia (e.g., diet, medication usage, physical activity). The current review clearly shows that only a few studies have simultaneously analyzed both the volatolome and gut microbiome profiles in colorectal neoplasia, and only one factored in diet. The question is whether VOCs arising in colorectal neoplasia are linked to disease pathophysiology or related to external risk factors such as diet. Previously, Hanna et al. proposed a framework for conducting and reporting future studies that investigate the role of VOCs in cancer diagnosis [135]. Future research should therefore focus on a multi-omics approach as understanding the role of different VOCs, their concentration, and their origin in the development of colorectal neoplasia is essential to acquire volatolome-based personalized diagnostic and treatment strategies. Furthermore, these studies should compare potential markers for colorectal neoplasia to those for other gut disorders, such as inflammatory bowel disease and irritable bowel syndrome. 

## 6. Limitations 

The current systematic review had some limitations. First, only very few papers simultaneously analyzed microbiota profiles in addition to VOCs. As a consequence, the majority of the results are to some extent extrapolated. Second, due to the heterogeneity in the methodological setup in metabolomics studies, caution is warranted in their interpretation, and reproduction and validation of findings is needed. By defining chemical or biological classes of interest, we have aimed to assess reproducible VOCs as objectively as possible, although it is possible that in this process important individual VOCs were missed due to missing classes of interest. Third, due to a limited number of studies investigating precursor lesions and the related heterogeneity in methodologies, common outcomes between these studies were hard to assess. Therefore, the volatolome for precursor lesions remains uncertain, although studies do pinpoint toward its early pre-cancer changes.

## 7. Conclusions

The gut microbiome contributes to the metabolic fingerprint of VOCs in patients with colorectal neoplasia. The exact role of the microbiome in shaping the host’s VOC profile in the detection of colorectal neoplasia is complex and remains to be further elucidated. Future research involving VOC analysis should aim for a multi-omics approach that also considers the gut microbiome and relevant external factors. 

## Figures and Tables

**Figure 1 metabolites-13-00055-f001:**
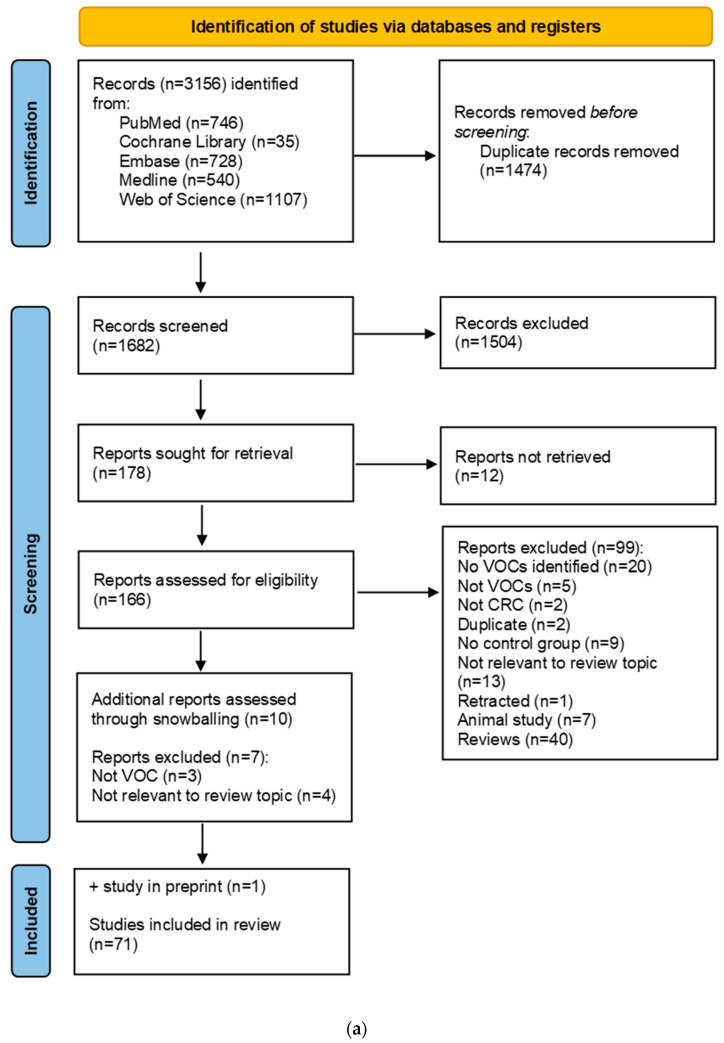
(**a**) PRISMA flowchart for search on VOCs and CRC. VOC: volatile organic compound, CRC: colorectal cancer. (**b**) PRISMA flowchart for systematic reviews and meta-analyses on CRC-related gut microbiota.

**Figure 2 metabolites-13-00055-f002:**
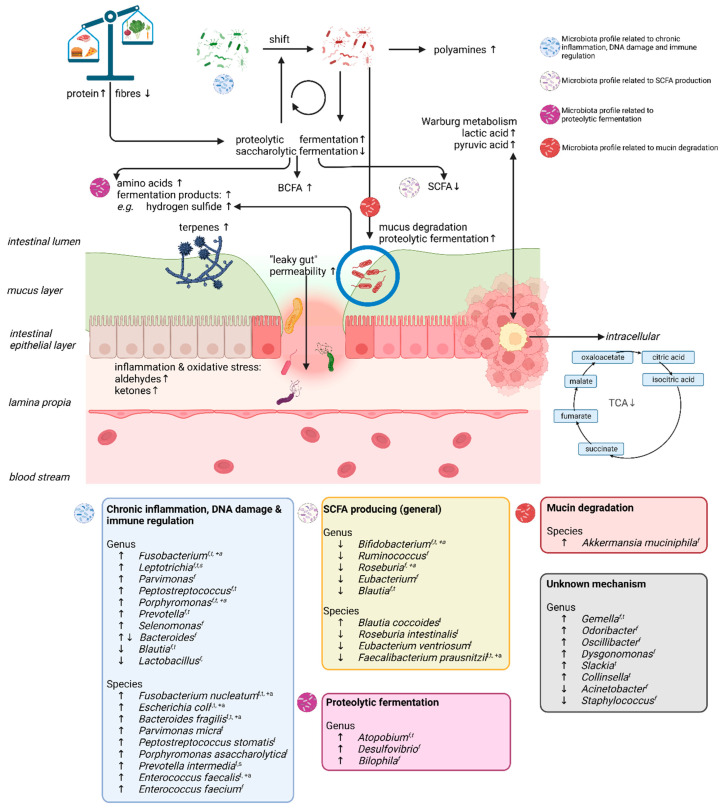
A schematic overview of the volatolome and gut microbial microenvironment in colorectal neoplasia. Diets low in fiber and rich in proteins cause a decrease in saccharolytic fermentation and an increase in proteolytic fermentation. As a result, detrimental amino acids and proteolytic fermentation products (including branched chain fatty acids) increase in concentration while beneficial short chain fatty acids decrease in concentration. This temporary effect can become long-term by adhering to such a diet for longer periods, presumably by causing shifts in gut microbiota. Unfavorable microbes can degrade host glycans and mucins in the intestinal lumen, which have a function in maintaining gut barrier integrity. This cycle is self-empowering. Moreover, such unfavorable gut microbiota shifts may include an unfavorable mycobiome, causing a rise in terpenes. As a consequence, in addition to other immune-related factors, the intestinal barrier function is compromised, causing low-grade systematic oxidative stress and giving rise to enhanced aldehyde and ketone concentrations. In turn, microbes may produce amines for protection. In addition, the microbiota may also be involved in pathways that enhance Warburg metabolism and thus empower cancerous cells by providing energy for fast growth, as the aerobic tricarboxylic acid cycle is downregulated in cancerous cells. In this figure, only the volatolome was considered in relation to the microbiota. CRC-associated microbiota data is adapted from references [100,101,102,103]. Up arrows denote CRC or adenoma-enriched VOCs and gut microbiota; down arrows denote CRC or adenoma-depleted VOCs and gut microbiota. BCFA: branched chain fatty acid, SCFA: short chain fatty acid, TCA: tricarboxylic acid cycle, VOCs: volatile organic compounds, f = fecal sample, t = tissue sample, s = saliva sample, +a = also observed in colorectal adenomas.

**Table 1 metabolites-13-00055-t001:** Relative VOC concentrations in CRC cases compared to control cases in feces, urine, breath, blood, tissue, and saliva.

Chemical Class	VOC	Feces	Urine	Breath	Blood	Tissue	Saliva
** *SCFA* **	acetic acid	C, n = 5		I, n = 2	I, n = 2		
	propanoic acid	C, n = 4		n = 1	I, n = 2		
	butyric acid	C, n = 7		n = 1	n = 1	n = 1	
	valeric acid	C, n = 2		n = 1	C, n = 2	I, n = 2	
	hexanoic acid	n = 1			n = 1		
	heptanoic acid				I, n = 2		
	octanoic acid			n = 1	I, n = 3	n = 1	
	nonanoic acid				C, n = 2		
	total SCFA	n = 1					
** *BCFA* **	isobutyric acid	I, n = 4		n = 1	I, n = 2		
	isovaleric acid	C, n = 3			I, n = 2	I, n = 2	
	ethyl 3-methylbutuanoate	n = 1					
	hexanoic acid 2-methyl		n = 1				
	4-hydroxybutyrate		n = 1				
	ethyl-acetate			n = 1			n = 1
	2-methyl butyric acid				C, n = 2		
	3-hydroxyisovaleric acid				n = 1		
	2-aminoisobutyric acid				n = 1		
	2-aminobutyric acid				I, n = 2	I, n = 2	
	2-hydroxybutanoic acid				C, n = 7		
	2-hydroxy-pentanoic acid						
	3-hydroxy butanoic acid				C, n = 6	n = 1	
	3-hydroxyisobutyric acid				n = 1		
	2-hydroxyisovaleric acid				n = 1		
	3-hydroxypropionic acid				n = 1		
** *Warburg metabolism* **	lactic acid	n = 1	n = 1	n = 1	C, n = 7	C, n = 5	
	pyruvate		n = 1		C, n = 5	n = 1	
	propyl-pyruvate			n = 1			
** *Amino acids* **	alanine	C, n = 4	n = 1		C, n = 4	C, n = 3	
	aspartic acid	C, n = 3	n = 1		I, n = 2	C, n = 4	
	valine	C, n = 4	C, n = 2		C, n = 8	C, n = 5	
	serine	C, n = 4	n = 1		n = 1	C, n = 3	
	proline	C, n = 3			C, n = 5	C, n = 7	
	glycine	C, n = 4	n = 1		C, n = 4	C, n = 6	
	phenylalanine	C, n = 2			C, n = 5	C, n = 6	
	leucine	C, n = 5	n = 1		C, n = 5	C, n = 4	
	threonine	n = 1			I, n = 2	C, n = 3	
	lysine	C, n = 2			C, n = 2	C, n = 3	
	isoleucine	C, n = 2	C, n = 2		C, n = 5	C, n = 5	
	citrulline	C, n = 1			C, n = 3		
	tryptophan		C, n = 2		C, n = 6	n = 1	
	histidine		I, n = 3		n = 1		
	tyrosine		n = 1		C, n = 7	C, n = 2	
	arginine		n = 1		C, n = 1	n = 1	
	ornithine				C, n = 5	n = 1	
	cysteine	n = 1			C, n = 5	C, n = 2	
	beta-alanine	n = 1			I, n = 3	C, n = 5	
	aspargine				I, n = 3	I, n = 2	
	glutamine				I, n = 2	C, n = 2	
** *Aldehydes* **	acetaldehyde	n = 1	n = 1	n = 1			n = 1
	heptanal	n = 1	n = 1	n = 1			
	hexanal		I, n = 3	n = 1			
	nonanal		n = 1	I, n = 3		n = 1	
	octanal		n = 1			n = 1	
	butanal		n = 1				
	undecanal		n = 1				
	decanal		n = 1	I, n = 4		n = 1	
	propanal			n = 1			
	pentanal			n = 1			
	benzaldehyde			I, n = 4		C, n = 2	
** *(branched) Alkanes/enes* **	heptane			n = 1			
	octane			n = 1			
	pentane		n = 1	n = 1			
	nonane				I, n = 2		
	decane			n = 1			
	dodecane			I, n = 4			
	tetradecane			I, n = 4			
	isoprene		n = 1	C, n = 2			
	4-methyl octane			n = 1		n = 1	
	2,2,1-dimethyldecane			n = 1			
	2-methyl-butane			I, n = 2			
	2-methylpentane			I, n = 2			
	3-methyl-pentane			n = 1			
	3-ethyl hexane				n = 1		
	5-butyl-nonane			n = 1			
	methylhexane			n = 1			
** *Sulfides* **	hydrogen sulfide	C, n = 2		n = 1			
	dimethylsulfide		n = 1				
	dimethyl disulfide		C, n = 2				
	carbondisulphide		n = 1				
	thiophene		n = 1				
	tetrasulfide, dimethyl					I, n = 2	
	dimethyl disulfide					I, n = 2	
	dimethyl trisulfide					n = 1	
** *Furans* **	2,4-dimethylfuran		n = 1				
	2-acetylfuran		n = 1				
	2-ethyl-5-methylfuran		n = 1				
	2-methyl-5-(methylthio)furan		n = 1				
	2-methyl furan		n = 1				
	2-pentyl furan		n = 1			n = 1	
	3-methylfuran		n = 1				
	furan		n = 1				
**Ketones**	hexan-2-one	n = 1					
	2-heptanone		n = 1			I, n = 2	
	2-pentanone		I, n = 2			n = 1	
	3-heptanone		n = 1				
	3-hexanone		n = 1				
	4-heptanone		n = 1				
	2,4-dimethyl-3-pentone		I, n = 2				
	3-methyl-2-butanone		n = 1				
	acetone		I, n = 2	C, n = 4			n = 1
	4-methyl-2-pentanone			I, n = 2			
	4-nonanone					n = 1	
	2-nonanone					I, n = 2	
** *Protein fermentation* **	ammonia	n = 1					
	tyramine	n = 1					
	p-cymene		n = 1				
	p-cresol		I, n = 3		n = 1		
	triethylamine			n = 1			
	trimethylamine			C, n = 2	n = 1		
	indole			n = 1		I, n = 2	
	phenol	n = 1	I, n = 2	n = 1		n = 1	
	phenyl lactic acid	C, n = 2					
	phenylacetic acid	n = 1					
	2,4-di-tert-butylphenol		n = 1				
	3,5-di-t-butylphenol		n = 1				
	4-methyl-phenol		n = 1				
	4-tert-butylphenol		n = 1				
	urea				C, n = 4		
	uric acid				C, n = 3	n = 1	
	taurine					I, n = 2	
	cadaverine	n = 1					
	indole acetate				n = 1		
	5-hydroxy-indoleacetate		n = 1				
	5-hydroxytryptophan		n = 1				
	putrescine		n = 1				
**Bacterial fermentation**	di-nitrogen oxide			n = 1			
	propanol						n = 1
	propan-2-ol	n = 1					n = 1
	propan-2-ul pentanoate	n = 1					
	propan-2-yl butanoate	n = 1					
	propan-2-yl propanoate	n = 1					
	ethanolamine	n = 1					
	methyl mercaptan	n = 1					n = 1
	ethanol			n = 1			n = 1
	choline				C, n = 2		
	norvaline	n = 1					
**Terpenes**	xylene	n = 1	n = 1				
	g-terpinene		n = 1				
	beta-pinene		n = 1	n = 1			
**TCA metabolism**	maltose	n = 1		n = 1	I, n = 2		
	fructose	C, n = 2				n = 1	
	malate	n = 1	n = 1		C, n = 3	C, n = 3	
	oxalic acid	n = 1	n = 1		n = 1	n = 1	
	succinic acid	C, n = 2	C, n = 3		n = 1	n = 1	
	citrate		C, n = 2		I, n = 4	n = 1	
	glucose		n = 1		I n = 2	C, n = 5	
	sorbose		n = 1		n = 1		
	xylose		n = 1		n = 1		
	arabitol		n = 1		n = 1		
	glucuronate		I, n = 2		n = 1		
	xylitol				I, n = 2		
	arabinose				I, n = 3	n = 1	
	isocitric acid		I, n = 2		I, n = 2		
	sucrose				n = 1		
	threitol				n = 1		
	fumaric acid	n = 1	n = 1		I, n = 2	n = 1	
	glutamic acid	n = 1	n = 1		I, n = 7	C, n = 5	
	glycolic acid				I, n = 2		
	D-mannose				C, n = 2	C, n = 4	
	ribitol				n = 1		
	ribulose				n = 1		
	glycerol				I, n = 2	C, n = 4	
	inositol				n = 1	C, n = 3	
	sorbitol					n = 1	

VOCs were grouped per relevant biological or chemical class. Colors represent the relative changes in CRC compared to control. Green, orange, and red signify the majority of studies reporting higher, inconclusive, or lower concentrations in colorectal neoplasia compared to controls, respectively. Inconclusive changes in concentrations (i.e., orange) can appear as result of concentration changes not being reported, as a result of contradicting observations, or due to combined effects. Similarly, green or red trends can be consistent or inconsistent, but with a clear trend for the majority. n = frequency of observations, C = consistent results, I = inconsistent results. C and I only reported where n > 1. SCFA: short chain fatty acid, BCFA: branched chain fatty acid, TCA: tricarboxylic acid cycle, VOC: volatile organic compound.

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
