# Peer review of "Systematic Review: Contribution of the Gut Microbiome to the Volatile Metabolic Fingerprint of Colorectal Neoplasia"

_metabolites, 2022, doi:10.3390/metabo13010055_

Round 1

Reviewer 1 Report

This review gives an excellent overview of how VOCs can be linked to gut microbiome applied to colorectal cancer. Many databases have been addressed to search relevant literature. The paper is well-written and is of great interest for both clinicians and for researchers in the field of volatomics. However, the authors performed two different searches, one regarding VOCs and CRC, and one concerning the microbial composition/activity and CRC. For the latter, the authors chose to include only systematic reviews as the literature was too extensive. The systematic reviews and the studies describing the VOCs are thus independent from each other and hence not determined in the same patients. So there is no direct link between VOCs and microbial composition/activity and the conclusions are speculative. Therefore, the 7 studies that did include both microbial composition and activity and VOCs should form the key publications and deserve a bit more attention as these are the only ones providing direct links between VOCs and microbiome.

Additionally, I have some suggestions that could improve the manuscript when integrating the answers into the paper.

·         Maybe you can add the different output of number of publications per database in PRISMA chart (fig 1a) and maybe you could change the direction of the arrows when using snowball referencing (now it seems that you are excluding studies instead of including them in your final set of papers)

·         Grammar should be checked as there were some typo’s.

·         The inclusion of in vitro cell line studies seems a bit odd regarding the goal of this review as this is a step further away from the clinical use. What was their added value? If not much, than maybe discard them from your search?

·         I understand the combination of VOCs into biological of chemical classes but this seems to nullify the reason of this review as general metabolic pathways of the microbiota (as mentioned by the authors in the review) or from tumorigenesis are known to liberate these classes. The added value thus lies within the linkage of specific VOCs to disease and gut microbiome and should therefore be the main focus and deserves more attention.

·         I was wondering if you have information on differences between precursor lesions and those with CRC in order to pinpoint potential screening biomarkers given that this is the main goal of correlating VOCs to CRC development?

·         Given that you want to use VOCs as markers for CRC, how certain are you that the volatiles that are described will be useful? More specifically, does the microbiome changes the colon environment and helps to develop CRC or does the transformation from precursor lesions to CRC induce a different environment and thereby a shift in microbial composition - and thus in VOCs - and will the cancer already be present before a shift in VOCs is expected (in which the added value of these markers will be low considering the cancer is already developed). I think this can be speculated more upon in the discussion. You give a nice example of lactate, but again, could this marker be useful (or specific enough) to detect CRC?

·         In addition to the previous remark, where are the markers different compared to e.g. IBD or IBS? Figure 2 also describes the leaky gut which is also key in those bowel diseases. How specific will your markers then be for CRC?

·         Can you speculate more in the review on the differences between matrices and the dynamics of VOCs and why some VOCs are seen in all matrices or only in some? How to explain why breath is no good matrix for CRC detection? Maybe the authors can add some advice on what kind of matrix should be investigated or used in the future? For instance, you explain why you see decreased concentrations in blood of amino acids, but do you also see this in feces or would you expect this to be seen there too? Why the discrepancy with other matrices?

·         Did you take into account potential environmental contaminants (e.g. from using sampling bags or desorption tubes) when listing the VOCs? Do you think terpenes are endogenous VOCs or rather as contaminants from food?

Reviewer 2 Report

The comments I can make based on my observation are:

-How did the authors check whether the same studies were included in different meta-analyses? If they have been checked and eliminated, then it is better to cite only the results of individual studies and not meta-analyses. Then meta-analyses were used to facilitate the extraction of papers. Pls. specify.

It is unclear what means sentence "Included studies were published between 2008 and 2022 and consisted of 71 original studies. The majority (n=68) analyzed..."    while in Figure 1b they state that they analyzed 5 studies only. Authors must clarify which claims throughout the text refer to studies and which referred to meta-analyses.

Figure 2 presents a very nice summary of their findings that is readable even by someone who is not a medical biochemist.
